# Barriers to Discharge in Geriatric Long Staying Inpatient and Emergency Department Admissions: A Descriptive Study

**DOI:** 10.3390/geriatrics6030078

**Published:** 2021-08-11

**Authors:** Kelsey J. Keverline, Steve J. Mow, Julianne Maire Cyr, Timothy Platts-Mills, Jane H. Brice

**Affiliations:** 1School of Medicine, University of North Carolina, Chapel Hill, NC 27514, USA; steve_mow@med.unc.edu; 2Department of Emergency Medicine, University of North Carolina, Chapel Hill, NC 27514, USA; jcyr@med.unc.edu (J.M.C.); tim_platts-mills@med.unc.edu (T.P.-M.); jane_brice@med.unc.edu (J.H.B.)

**Keywords:** healthcare utilization, social determinants of health, barriers to discharge, boarding, length of stay, emergency medicine, inpatient care, geriatrics

## Abstract

Background: This study describes long length of stay during emergency department (ED) visits and hospital admissions, barriers to discharge, and discharge solutions for geriatric patients. Methods: We conducted a retrospective medical record review of a random sample of 150 ED patients and 150 inpatients with long length of stay (LOS) encounters. Cohorts were characterized by demographics, social determinants of health (e.g., health insurance, housing), medical comorbidities at admission, discharge care coordination, and final disposition. Results: In the ED, the primary barrier to discharge was inadequate inpatient bed availability (63%). In the inpatient setting, barriers to discharge were predominantly due to a demonstrated medical requirement for continued hospitalization (55%), followed by difficulty with coordinating discharge to a skilled nursing facility or rehabilitation center (22%). Discussion: Among long LOS ED patients, discharge delays were often the result of unavailable inpatient beds and services. Reducing the LOS for ED patients may require further investigation as to which hospital services are most frequently utilized by geriatric patients and structuring inpatient bed allocation to prevent extended patient boarding in the ED. Reducing long inpatient LOS may require early identification of high-risk patients and strengthening of relationships with community-based services.

## 1. Introduction

As the population of individuals aged ≥65 years grows in the United States (U.S.), demand for healthcare services rises concordantly. In 2019, the United States Census Bureau estimated persons ≥65 years represented 16.5% of the total population [1]. As the U.S. population continues to age, the demand for healthcare services providing care to geriatrics increases. For example, in 2017, for every 100 persons aged ≥65 years, 45 emergency department (ED) visits were recorded annually compared to 35 ED visits for persons aged 45–64 [2]. ED visits for older adults are more likely to result in unnecessary inpatient admissions compared to visits for younger patients [3].

Emergency departments nationally have reported escalating rates of prolonged boarding of admitted patients [4]. Boarding is a hospital-based problem indicating insufficient supply to meet healthcare demands. Some have postulated that boarding may reflect prolonged inpatient length of stay (LOS) secondary to social determinants of health (SDOH) and perhaps due to insufficient outpatient or long-term care resources [5]. Prolonged LOS in the ED has been found to result in increased medical errors, increased in-hospital LOS and mortality [6], increases in left without being seen rates, and EMS diversions, which can cause potential delays in essential care [7,8]. Similarly, prolonged inpatient length of stay has been found to result in an increase in hospital acquired infections, deep vein thrombosis, and increased in-hospital mortality [9]. In addition to health consequences, long LOS (LLOS) leads to increased healthcare costs and lack of availability of inpatient resources for other patients. A previous study at a university teaching hospital identified that 13.5% of bed-days out of 2831 in an academic medical center were unnecessary for acute inpatient care [10].

SDOH, including factors outside of received medical care and genetics which influence a person’s health and wellbeing, may prolong the LOS for some patients. Most prior studies in the geriatric population have focused on clinical predictors of LLOS rather than SDOH [11,12]. Studies from Singapore and France have suggested caregiver stress and difficulties in nursing home placement were risk factors for LLOS [13,14]. The population demographics and healthcare structures in these countries are very different from the United States making it difficult to generalize these findings. 

We sought to identify barriers to discharge (BTDs) for geriatric LLOS patients in ED and inpatient settings and to differentiate LLOS encounters by medical versus non-medical BTDs. Further, we sought to understand the demographics of LLOS patients and the solutions to discharge delays. 

## 2. Methods

### 2.1. Study Setting

This study was a retrospective chart review of 150 ED and 150 inpatient LLOS encounters randomly selected from all ED and inpatient visits at the University of North Carolina Hospitals in 2018. The University of North Carolina Hospitals is a Level 1 trauma center and academic hospital in the southeastern United States. The study was approved by the Institutional Review Board (IRB) at the participating institution. 

### 2.2. Inclusion Criteria

Patients were considered eligible for the study if they were ≥65 years of age at the time of patient encounter. Patient encounters were included if they had an ED length of stay greater than 24 h (96th percentile of the ED’s LOS) or an inpatient length of stay of at least 20 days (Center for Medicare and Medicaid Services definition, 2016). Any encounter occurring at least in part during the year 2018 was eligible for inclusion. All patient encounters meeting inclusion criteria were included regardless of reason for care, ensuring a real-world review of patient encounters. Incarcerated individuals were excluded from the study. 

### 2.3. Data Sources and Collection

All inpatient hospitalizations which occurred at least in part during the year 2018 were identified by Carolina Data Warehouse for Health (CDWH), the research data repository associated with the University of North Carolina Chapel Hill health center. An IRB-approved study was submitted to CDWH detailing inclusion and exclusion criteria for inpatient encounters. The chief clinician and principal investigator of the study used ED electronic medical records (EMR) to identify all ED encounters from the year 2018 with LOS > 24 h. 

Each encounter identified was given a deidentified study ID. Encounters were then randomized using a random number generator in Microsoft Excel. For this pilot project, a random sample of 150 inpatient and 150 ED encounters were obtained from these populations. All charts were reviewed and data were documented by one team member, as described below. 

Patient demographics (e.g., age, race, sex), SDOH (e.g., health insurance status, housing status, immigration status), reason for care (i.e., chief complaint in ED or primary diagnosis on discharge paperwork from the inpatient setting), nature of the BTD (i.e., medical reasons or non-medical reasons), social reasons for delayed discharge (e.g., lack of appropriate discharge site, coordination with patient and/or family, financial barriers), ultimate discharge solutions (DS), and discharge disposition (DD) were identified from charts. 

The framework described by Ravigan et al. (2017) was used to identify and organize BTD and DS for ED and inpatient hospitalizations [15]. BTDs were identified as medical if the patient’s condition necessitated hospital-level medical care for the full LOS. BTDs were considered social if a barrier prevented discharge while the patient was medically cleared for care at a lower level of care (home health services, home with caregiver) or the patient was awaiting transfer to another facility (e.g., skilled nursing facility, assisted living facility). As previously identified, three barriers were specific to the ED which were not applicable to the inpatient hospital setting; these included active suicidal ideation, active homicidal ideation, and lack of inpatient hospital bed availability. BTDs were determined as “primary” or “secondary” depending on factors determined to be most prohibitive to discharge. All patients had an identified primary BTD, but not all patients had an identified secondary BTD. 

Total LOS for patient encounters in the ED and inpatient setting were recorded. For inpatient encounters, the day at which a patient was deemed medically cleared for discharge by the care team was identified and then the encounters were characterized by the fraction of inpatient stay that was medically necessitated as compared to a SDOH-related delay. This determination was made using medical provider notes, care manager notes, and social worker notes. Because of the nature of ED charts, it was indeterminable through chart review when a patient was “medically cleared” for discharge. 

Descriptive statistics as outlined above were calculated in Microsoft Excel version 2012. Demographics results from LLOS ED and inpatient encounters, were then compared using chi-square tests (α = 0.05).

## 3. Results

The calendar year of 2018 contained 368 ED and 564 inpatient geriatric LLOS encounters. These visits accounted for 0.64% of all ED encounters (56,917 total ED encounters) and 0.13% of all inpatient encounters (446,410 total inpatient encounters) at the study institution. Within the sample, the median ED LOS was 37.7 h (range 24.0 to 1197.1 h) and the median inpatient LOS was 28.2 days (range 20.0 to 221.9 days). When stratified by psychiatric vs. non-psychiatric reason for admission, the median ED psychiatric LOS was 58.8 h (range 24.2 h to 1197.1 h) and the median ED non-psychiatric LOS was 33.12 h (range 24.0 h to 92.7 h). For inpatient LLOS encounters, the median inpatient psychiatric LOS was 35.5 days (range 21.9 to 77.7) and the median inpatient non-psychiatric LOS was 27.7 (range 20.0 to 221.9) days. Total bed days occupied by ED LLOS patients observed in the study were 430 bed days and total bed days occupied by inpatient LLOS patients were 5519 bed days. This encompasses 2% of all ED inpatient bed days (21,543 total bed days) and 2% of all inpatient bed days (254,763 total bed days) just for the 300 patients observed in the study. If we take the sample of 150 ED and 150 inpatient encounters as an appropriate sample of geriatric LLOS encounters, we can extrapolate the average bed days per patient to 1054 ED bed days (5% of total bed days taken up by 0.64% of ED patients) and 20,751 inpatient bed days (8% of total bed days taken up by 0.13% of inpatient patients). 

Table 1 provides comparisons of ED and inpatient patient demographics. There are statistically significant differences between the two populations in terms of age (*p* = 0.0090), race (*p* = 0.0051), insurance status (*p* = 0.019), and housing status (*p* = 0.0005). Patients with inpatient LLOS were more likely to be younger geriatric patients (age range 65–74), while patients with ED LLOS were more likely to be older geriatric patients (age ranged 85–95). Patients with inpatient LLOS were more likely to be Black/African American, Asian, or “Other” while patients with ED LLOS were more likely to be White. We found that patients with ED LLOS were more likely to have Medicaid or private insurance as the primary insurance while patients with inpatient LLOS were more likely to have Medicare or Medicare Advantage programs as the primary insurance. Finally, we found that patients with ED LLOS were much more likely to arrive from assisted living facilities or skilled nursing facilities while patients with inpatient LLOS were more likely to have had an independent residence prior to hospitalization. Chi-squared analysis of the samples showed no significant differences between the two patient populations in terms of sex, ethnicity, or primary language spoken (*p* > 0.05). 

Table 2 explores primary illness categories of ED and inpatient LLOS patients. The top three illness categories among ED geriatric LLOS patients were psychiatric (51%) followed by cardiovascular (15%) and communicable illnesses (10%). The top three illness categories among inpatient geriatric LLOS patients were cardiovascular (23%), oncologic (21%), and unintentional traumatic injuries (11%). 

Table 3 provides comparisons of the BTD and DS for ED and inpatient LLOS encounters. The largest BTD faced by ED patients was a lack of available hospital beds (65%). The most common DSs for ED patients were thus the availability of an inpatient hospital bed (55%) or transfer to another facility (21%). For inpatients, the most common BTD was medical necessity for continued hospitalization (57%) and the most common DS was resolved medical necessity (49%). The most common SDOH BTD and DS in the inpatient population was discharge site coordination (22% and 23%, respectively). Secondary BTDs were also identified in cases where applicable. Eighty-six ED encounters had a secondary BTD identified, of which the most common were due to discharge site coordination (50%). Thirty-six inpatient encounters had a secondary BTD identified, of which the most common was availability of follow up care (25%)

Table 3 also describes the disposition of LLOS patients. Most LLOS ED patients were ultimately admitted to the hospital (61%). Most LLOS inpatients ultimately required further medical care upon discharge. More than one-third (37%) of inpatient patients were discharged to long term acute care facilities or skilled nursing facilities, while 27% of patients were able to be discharged home with home health services in place.

Fifty-six inpatients’ stays (37%) were extended at least in part by SDOH as determined by care team and case manager notes. Of these, the average stay was 17.5 days after the patient was deemed medically cleared for discharge. 

## 4. Discussion

In this academic medical center, patients with LLOS often faced BTDs that were not exclusively medical in nature, and SDOH played a role in prolonging LOS for these patients. Among LLOS ED patients, discharge delays were often the result of unavailable inpatient beds and services. The primary BTD for ED patients was limited bed availability inside the hospital. In the inpatient setting, a majority of LLOS patients experienced prolonged hospitalization due to medical necessity. When BTDs arose, they were often related to safe discharge coordination, whether that included alternative housing or initiation of in-home resources. Secondary barriers to discharge, when identified were predominantly non-medical in nature. The identification of secondary barriers to discharge speaks to the reality that many delays in discharge are multifactorial in nature and may be related to interconnected medical and SDOH reasons. 

One of the largest differences between LLOS patients in the ED and inpatient settings is the discrepancy in LOS for psychiatric patients. Psychiatric patients were included in this LOS study to accurately characterize the reality of the ED and inpatient settings. In the ED, psychiatric patients had a median LOS almost double the LOS for non-psychiatric patients. In addition, the range of stay for psychiatric patients was seventeen times greater than the range of stay for nonpsychiatric patients. This likely relates to the primary BTD being lack of inpatient beds and the limited psychiatric beds available within and outside of the study center; this also indicates a higher need for geriatric psychiatric resources than are currently available. In the inpatient setting, non-psychiatric patients had a higher median length of stay and a greater range of length of stay. This suggests that while psychiatric patients may not necessarily require longer hospitalizations than non-psychiatric patients, there are significantly fewer inpatient resources for geriatric psychiatric patients than for non-psychiatric patients. 

Demographically, patients with inpatient LLOS have been previously identified as younger, minority males. When examining only geriatric LLOS patients, minority patients were more likely to have LLOS in the inpatient setting while White patients were more likely to have LLOS in the ED. This reaffirms previous research indicating minority patients make up a higher percentage of inpatient LLOS. In addition, our study found that among geriatric patients, inpatient LLOS patients were more likely to belong to a younger age bracket (65–74) than the ED LLOS patients who were more likely to belong to the oldest age bracket (85–95). This difference might reflect a difference in the housing status of those in the younger age bracket (65–74) and those in the oldest age bracket (85–95). Those in the oldest age bracket are more likely to live in an assisted living facility or skilled nursing facility setting. Since the care facility is responsible for the well-being of that resident, they are more likely to send the resident to the ED if there is a concern about their health. For recently hospitalized adults who are transferred to skilled nursing facilities, 22% end up requiring additional ED or hospital care within thirty days [16]. This is also supported by anecdotal reports from providers and EMS personnel who report frequent use of EMS calls to skilled and nonskilled facilities due to concerns for a resident’s wellbeing. Individuals who are living independently may be more likely to avoid hospital services. These findings support previous work of van den Broek et al. who found that preventable hospital admissions in older adults are often related to postponement of medical care in patients arriving from an independent living setting [3]. Previous studies have shown that residents of assisted living facilities and skilled nursing facilities have higher hospital admission rates and ED visit rates than community dwelling older adults however, the findings from this study indicate that community dwelling older adults, once hospitalized, are more likely to experience LLOS [17]. 

Geriatric patients who experienced inpatient LLOS were more likely to have Medicare or Medicare advantage insurance coverage while geriatric patients who experienced LLOS in the ED setting were more likely to have Medicaid or private insurance. Previous research has shown that individuals who experience LLOS are more likely to be uninsured or to have public insurance programs. Although many in the study population did have public health insurance, only four total patients in our study were uninsured, due to Medicare coverage of most geriatric patients. 

Inpatient LLOS patients were more likely to be live independently prior to hospitalization while ED LLOS patients were more likely to live in assisted living facilities or skilled nursing facilities. The higher proportion of inpatient LLOS patients coming from independent living situations could reflect that a large BTD on the inpatient setting is appropriate facility placement after a significant medical event or rehabilitation of the patient until they are able to return to an independent living setting or discharge to a lower level of care. 

## 5. Limitations

This study collected data from a single center in the southeastern U.S. that is a safety net institution. In this way, results of this study may not be generalizable, especially to hospitals that are not academic, urban, safety net institutions. 

In many instances, the point at which a patient is medically stable for discharge and requires continued hospitalization for social reasons is not fully documented in the patient’s chart. This is particularly a concern in the ED, where case managers are not assigned to every patient and it is unclear why some patients remain in the ED over 24 h. Furthermore, medical-system barriers to discharge are suspected to be underreported in patient’s charts. These include delayed physician orders, unavailability of hospital services over the weekend, and social worker availability. 

Our study initially sought SDOH data including immigration status and housing status, but this information was not routinely collected and documented in patients’ charts. One limitation in identifying if immigration status contributes to LLOS is the inherent risk of patient-provider discussion of immigration status. When providers ask questions about immigration status, this can damage the therapeutic alliance over fear of retaliation or deportation. Another limitation impacting the interpretation of housing status’s impact on LLOS is lack of information about family supports in the area. The decision for an older adult to maintain independent living or move to an assisted living facility or skilled nursing facility depends on the availability of family members to aid with care after a significant medical event. Because caregiver information is not reliably recorded in the medical record, a prospective longitudinal study may be more appropriate to understand this facet of LLOS. 

## 6. Conclusions

Our work continues to demonstrate that many barriers to discharge in the ED setting are due to lack of available inpatient beds for patients. Furthermore, we identified that geriatric patients with psychiatric needs encountered increased delays in appropriate treatment compared to geriatric patients with non-psychiatric needs. Reducing LLOS in the ED may require further investigation into which hospital services are most frequently required by geriatric patients. In the inpatient setting, we found that geriatric patients experienced significant delays in discharge due to facility placement issues. Thus, reducing inpatient LLOS may require early identification of high-risk patients and strengthening relationships with community-based services. 

## Figures and Tables

**Table 1 geriatrics-06-00078-t001:** Demographic Makeup of ED and inpatient LLOS patients.

	ED LLOS *n* (%)	Inpatient LLOS *n* (%)	
Sex			*p* = 0.106
Female	84 (56%)	70 (47%)	
Male	66 (44%)	80 (53%)	
Primary Language			*p* = 0.217
English	149 (99%)	145 (96%)	
Spanish	1 (1%)	3 (2%)	
Other or Unknown	0 (0%)	2 (1%)	
Housing Status			*p* = 0.001
Permanent Independent Housing	114 (76%)	141 (94%)	
Assisted Living Facility	18 (12%)	3 (2%)	
Homeless	0 (0%)	0 (0%)	
Sheltered Temporarily	5 (3%)	0 (0%)	
Skilled Nursing Facility	13 (9%)	4 (3%)	
Group Home	0 (0%)	0 (0%)	
Other or Unknown	0 (0%)	1 (1%)	
Insurance Status			*p* = 0.019
Medicare	57 (38%)	66 (44%)	
Medicaid	6 (4%)	2 (1%)	
Private Insurance	16 (11%)	5 (3%)	
Medicare Advantage	64 (42%)	75 (50%)	
Charity Care	3 (2%)	2 (1%)	
Unknown	4 (3%)	0 (0%)	
Race			*p* = 0.005
White	115 (77%)	89 (60%)	
Black or African American	32 (21%)	48 (32%)	
Asian	0 (0%)	2 (1%)	
American Indian or Alaskan Native	3 (2%)	2 (1%)	
Other	0 (0%)	6 (4%)	
Unknown	0 (0%)	2 (1%)	
Ethnicity			*p* = 0.153
Hispanic or Latinx	2 (1%)	4 (3%)	
Non-Hispanic or Latinx	148 (99%)	143 (96%)	
Unknown	0 (0%)	3 (2%)	
Age			*p* = 0.009
65–74	89 (59%)	105 (70%)	
75–84	44 (29%)	42 (28%)	
85+	17 (11%)	4 (3%)	

ED, emergency department; LLOS, long length of stay; LOS, length of stay.

**Table 2 geriatrics-06-00078-t002:** Primary Illness Categories Identified for ED and Inpatient LLOS Encounters.

	ED *n* (%)	Inpatient *n* (%)
Mental and Behavioral Disorders	76 (51%)	14 (9%)
Cardiovascular and Circulatory Diseases	23 (15%)	35 (23%)
Cancer	8 (5%)	31 (21%)
Communicable Illnesses	15 (10%)	13 (9%)
Unintentional Traumatic Injuries	5 (3%)	16 (11%)
Digestive Diseases	5 (3%)	13 (9%)
Chronic Respiratory Diseases	6 (4%)	8 (5%)
Diabetes/Urogenital/Blood/Endocrine	4 (3%)	8 (5%)
Neurological Disorders	3 (2%)	7 (5%)
Other Noncommunicable	4 (3%)	2 (1%)
Musculoskeletal Disorders	0 (0%)	3 (2%)
Intentional Injuries	1 (1%)	0 (0%)

**Table 3 geriatrics-06-00078-t003:** Barriers, Solutions, and Results of LLOS Patient Discharge.

	ED *n* (%)	Inpatient *n* (%)
Primary Barrier to Discharge		
Medical Necessity	9 (6%)	85 (57%)
Medical Barrier	23 (15%)	13 (9%)
Discharge Planning	4 (3%)	7 (5%)
Discharge Site Coordination *	12 (8%)	33 (22%)
Coordination with Patient and/or Family *	4 (3%)	6 (4%)
Financial Reason *	0 (0%)	1 (1%)
Hospital Bed Availability **	97 (65%)	--
Suicidal Ideation **	0 (0%)	--
Other/Unable to Obtain	1 (1%)	0 (0%)
Discharge Solution		
Medical Necessity Resolved	7 (5%)	74 (49%)
Deceased	0 (0%)	12 (8%)
Medical Barrier	17 (11%)	10 (7%)
Discharge Planning *	5 (3%)	9 (6%)
Discharge Site Coordination *	31 (21%)	35 (23%)
Coordination with Patient and/or Family *	5 (3%)	6 (4%)
Financial Reason *	0 (0%)	0 (0%)
Hospital Bed Availability **	83 (55%)	--
Suicidal Ideation **	0 (0%)	--
Other/Unable to Obtain	2 (1%)	0 (0%)
Disposition		
Admitted Inpatient ***	92 (61%)	--
Other Healthcare Facility	21 (14%)	5 (3%)
Hospice	1 (1%)	10 (7%)
Home	22 (15%)	26 (17%)
Home with Home Health	3 (2%)	40 (27%)
Skilled Nursing Facility or Post-Acute Care Facility	4 (3%)	56 (37%)
Left Against Medical Advice	1 (1%)	0 (0%)
Other	0 (0%)	1 (1%)
Deceased	0 (0%)	12 (8%)

* Barrier considered social determinant of health. ** Barrier to discharge specific to ED. *** ED specific disposition.

## Data Availability

Data supporting this research was obtained from the Carolina Data Warehouse and EPIC chart review.

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
