# Peer review of "Barriers to Discharge in Geriatric Long Staying Inpatient and Emergency Department Admissions: A Descriptive Study"

_geriatrics, 2021, doi:10.3390/geriatrics6030078_

Round 1

Reviewer 1 Report

  1. In the abstract, please use full phrase for LOS when using it for the first time.
  2. Line 29-32: the authors are wanting to describe increase needs for geriatric care as the age increases, but in the example there is no comparison with previous years or different age group.
  3. Please provide citation for line 37-39.
  4. Line 46 – 13.5% of bed-days over what period of time?
  5. Did the authors exclude patients who had LLOS in the ED and subsequently became inpatient and also had LLOS to avoid potential duplication?
  6. The results are somewhat skewed due to inclusion of psychiatric patients in the ED encounters.
  7. Due to a single center retrospective study, results are not generalizable. 
  8. This is overall good study but does not provide any new details. These are well known BTD.  

Author Response

Firstly, we would like to thank you for your time and consideration of our manuscript.  We look forward to using your feedback to improve our manuscript.  

1) In the abstract, please use full phrase for LOS when using it for the first time. ​

This has been addressed. Thank you!

2) Line 29-32: the authors are wanting to describe increase needs for geriatric care as the age increases, but in the example there is no comparison with previous years or different age group. 

We have updated the example to provide a comparison with a different age group

3) Please provide citation for line 37-39

.This has been addressed, thank you!

4) Line 46 – 13.5% of bed-days over what period of time?

This has been addressed. Thank you!

5) Did the authors exclude patients who had LLOS in the ED and subsequently became inpatient and also had LLOS to avoid potential duplication?

​We did not choose to exclude these patients in our study.  While there might be duplication of patients, we thought that the barriers to discharge in the ED and inpatient setting were sufficiently different to provide novel information or insight into how SDOH contribute to BTDs. If duplication has happened in the patient population studied, it is thought to be rare.  

6) The results are somewhat skewed due to inclusion of psychiatric patients in the ED encounters.

​Our goal in this study was to provide a snapshot of the ED "as is."  The Emergency Department does provide boarding for many psychiatric patients for an extended period.  We discuss in the results section the difference in the median and range of psychiatric and non-psychiatric visits. In addition, psychiatric patients are often excluded from analysis such as ours which our team noted on our review of literature.  This exclusion does not adequately represent the problems of ED boarding.

7) Due to a single center retrospective study, results are not generalizable.

​Our hope is to expand the research in the future to include smaller centers as well as more geographically diverse centers. This is a snapshot of a problem and forms the basis of our team's further work to understand this problem.  Additionally we have cited literature in the manuscript that suggests this is a problem that occurs nationally. 

8) This is overall good study but does not provide any new details. These are well known BTD.  

Thank you for this comment. While these BTD are well-known, little has been done to correct or mitigate them. IT is our hope that by shining a new light on them, we might frame the conversation and drive the creation of new solutions. This is the strong desire of our team.

Reviewer 2 Report

The results of the study are important and will be of use elsewhere. However, in the Discussion the information given in the results section is dense and therefore difficult to wade through.  Would it be possible to display the statistically significant factors in a figure (or a table)? The figure could include BTDs, consequences of LLOS, significant SDOH factors; illness categories.  A side-by-side comparison of statistically significant results of long length of stay of a ED visit  vs long length of stay as an inpatient would make the results clearer. The figure would replace much of the verbiage in the Discussion, thereby shortening the manuscript.

line 131-132: how did you choose 150 as an appropriate sample size?

You use both healthcare and health care. Be consistent.

Additional key words to consider adding include length of stay, inpatient care, geriatrics, barriers to discharge, boarding

Author Response

Firstly, we would like to thank you for your time and consideration of our manuscript.  We look forward to using your feedback to improve our manuscript.

1) The results of the study are important and will be of use elsewhere. However, in the Discussion the information given in the results section is dense and therefore difficult to wade through.  Would it be possible to display the statistically significant factors in a figure (or a table)? The figure could include BTDs, consequences of LLOS, significant SDOH factors; illness categories.  A side-by-side comparison of statistically significant results of long length of stay of a ED visit  vs long length of stay as an inpatient would make the results clearer. The figure would replace much of the verbiage in the Discussion, thereby shortening the manuscript.

We have added an additional table and combined tables 2 and 3 to address this comment.  We hope the results make the results section more streamlined and reader friendly.  In addressing statistical significance, for many of our categories we could not directly compare the ED and inpatient populations given that the BTDs, solutions to discharge, and dispositions differ between the two populations.  For example, we did not feel like we could make a statistically significant comparison of the ED and inpatient BTDs given that the largest BTD in the ED setting is awaiting hospital admission.  

2) line 131-132: how did you choose 150 as an appropriate sample size?

This was a pilot study for the purpose of identifying BTD among the ED and inpatient setting using a previously developed schema by Ravagan et al. This study was not statically powered, but rather designed as a snapshot of cases that could be assessed by a small research team.

3) You use both healthcare and health care. Be consistent.

​This has been addressed. Thank you!

4) Additional key words to consider adding include length of stay, inpatient care, geriatrics, barriers to discharge, boarding

This has been addressed. Thank you!

Reviewer 3 Report

Thanks for the opportunity to read this paper. I found it to be very well written and conducted. The findings are interest and relevant to clinicians and management. Overall the quality is very high.

Author Response

Thank you so much for your time and feedback!